# Changes in Rainfall in Sierra Leone: 1981–2018

**Richard Wadsworth *** , **Amie Jalloh and Aiah Lebbie**

Department of Biological Sciences, Njala University, Njala, PMB Freetown, Sierra Leone;
amie_jalloh91@yahoo.com (A.J.); alebbie@njala.edu.sl (A.L.)

*** Correspondence: rwadsworth@njala.edu.sl; Tel.: +232-79675004

**Abstract:** Sierra Leone on the west coast of Africa has a monsoon-type climate. Reports by politically influential donors regularly state that Sierra Leone is extremely vulnerable to climate change, but the objective evidence backing these statements is often unreported. Predicting the future climate depends on modelling the West African monsoon; unfortunately, current models give conflicting results. Instead, changes in rainfall over the last four decades are examined to see if there are already significant changes. Rainfall records are extremely limited, so the Climate Hazards Group InfraRed Precipitation with Station daily data at a spatial resolution of 0.05 degrees was used. In addition to total annual rainfall, the characteristics of the early rainy season (critical for farmers), the length of the rainy season and growing season, and the frequency of extreme events were calculated. There is evidence for a significant reduction in annual rainfall in the northwest. There is only limited support for the widely held belief that the start of the rainy season is becoming more erratic and that extreme events are becoming more common. El-Niño was significant in the southeast. If these trends continue, they will exacerbate the consequences of temperature increases (predicted to be between 1 and 2.6 °C by 2060) and negatively affect the livelihoods and agricultural practices of the rural poor.

**Keywords:** Sierra Leone; rainfall; climate change; CHIRPS; El Niño; livelihoods

## 1. Introduction

While the debate over climate change may seem to be a relatively modern issue, concern over decreasing rainfall in Africa is not a new phenomenon. In the past, it has often been based on anecdotal and unverified data; this paper uses a rigorous data-driven approach to determine whether the rainfall pattern in Sierra Leone is changing. Sierra Leone has an unfortunate history of civil strife; some authorities [1] attribute this to conflict over natural resources. While a direct link between climate change and political instability is refuted [2], more nuanced assessments show that adverse effects on the economy that can lead to instability are plausible [3]. Given the recent history of Sierra Leone, it is important that the consequences of changes in rainfall might have on agriculture and the environment (and hence the economy) is understood.

In the 19th century, explorers and missionaries were convinced that Southern Africa was drying up. In 1849 David Livingstone noted, "Other parts 14 miles below the Kuruman gardens are pointed out as having contained, within the memory of people now living, hippopotami and pools sufficient to drown both men and cattle" [4]. An analysis of nearly 2000 letters of missionaries and reports by Enfield and Nash [4] suggests that, between 1815 and 1900, there were six droughts and seven wet periods. Early Missionaries met droughts, and the later ones continued the narrative of desiccation established by their predecessors and ignored the wet periods. In the 20th century, attention switched to the Sahel region, where there were major droughts in the 1970s and observations of changes in rainfall over decadal time scales are necessary [5].

In Sierra Leone, agriculture contributes about half of GDP and is the major source of income for the rural population [6]. Most agriculture is carried out by small farmers, using traditional approaches,

particularly "bush fallow" (allowing natural revegetation for 5 to 10 years after each cropping period). Almost all agriculture is rain-fed, and there is little modern irrigation. Like the 19th -century missionaries, various donors and decision makers are convinced that they already know how the climate is changing and that the country is extremely vulnerable to climate change: "Climate Change is known to have adversely affected the environment. . . . Flooding is known to have affected agriculture and habitats of people in Sierra Leone and their suffering aggravated by the attending health problems of water-borne diseases (typhoid dysentery cholera and diarrhoea)." (emphasis added) [7] or that "Climate change should be included in the mandate of the Disaster Management Department, ONS (office of National security)" [8]. The consultants Maplecroft rank Sierra Leone as the third most vulnerable to the economic impacts of climate change (behind Bangladesh and Guinea-Bissau) [9]. Unfortunately, predicting the effect of climate change on the West African Monsoon (WAM) is difficult. A project, CORDEX (Coordinated Regional climate Downscaling Experiment) [10] attempted to assess multiple regional climate models, and, in verifying them, given the paucity of good ground observations, they conclude, "Finally, we would emphasize that the ensemble average should not be viewed as the expected outcome in CORDEX-generated climate projections". In neighboring Liberia, eleven climate models were run for three sites, generating predictions varying from –36% to + 21% (Monrovia), –40% to + 24% (Nimba), and –40% to + 35% (Sapo National Park); Stanturf et. al. [11] conclude "predictions of precipitation in Liberia lack any sense of consistency". The situation in neighboring Guinea is also characterized by limited data; Aguilar et. al. [12] reported eight long-term rainfall records, but used only four in their assessment of extreme events, showing a reduction in intensity and total precipitation, but New et al. [13], in Gambia, identified increases in extreme rainfall, and especially in the daily intensity of precipitation (from two meteorological stations). Predicted changes in rainfall in Sierra Leone provided by the World Bank from a large suite of models are in the range of approximately plus or minus 100 mm, with a median estimate of a small increase [14].

For a variety of reasons, there are few reliable long-term rainfall records in Sierra Leone. The WMO (World Meteorological Organization) provides monthly average rainfall for five sites for the period 1961–1990 [15]. Statistics Sierra Leone [16] reports monthly rainfall totals for four sites for seven years (2007–2013); there is no statistically significant correlation between the annual totals for any pair of sites. An analysis of monthly rainfall records in Freetown from 1960 to 2009 found large fluctuations on decadal timescales [17]. Increases were attributed to a mixture of increased convection (more buildings leading to higher temperatures in the city) and increased condensation nuclei (dust and smoke particles), but their two graphs are difficult to reconcile. Four rainfall records (Freetown, Lungi, Bo, and Daru) were identified for the period 1990–2014, and they showed changes in rainfall in terms of the onset and duration of the rainy season and a correlation between annual total and length of the rainy season [18]. Data availability was much better in the past; Hayward and Clarke [19] analyzed records for 31 rain gauges in the northern part of the Freetown Peninsular (an area of only about 15 km by 15 km). They wanted to assess the effect of elevation and rain shadow on annual rainfall; however, the data predates 1970, and more than half the gages (17) were at very high elevation in relation to the Guma reservoir.

## 2. Objective

The objective of this research is to achieve the following;

- Quantify trends in rainfall over the past few decades and determine if the trends are statistically significant.
- Examine the spatial patterns in rainfall at a resolution suitable for planning.

As well as changes in total annual rainfall, the following measures of rainfall that have agricultural and environmental importance are calculated:

- Rainfall during the planting season, (mid-March to the end of May);

- Start date of the main rainy season;
- How erratic the start of the rainy season is;
- Length of the rainy season and growing season;
- Extreme events, as the annual maximum one-day and seven-day rainfall totals.

## 3. Method

A detailed comparison of 26 global precipitation records produced from satellite data, gauge data, and a combination of satellite and gauge data has recently been completed [20]. The analysis was performed by using data from North America, although they recommend a similar analysis be undertaken in other areas, such as Australia. For the purpose of considering climate change, a long record is essential, and planning a high spatial resolution is preferable. The CHIRPS (Climate Hazards Group InfraRed Precipitation with Station data) gridded data set [21] is produced by a synthesis of satellite and gauge-based data. It was assessed [20] as having low bias and variability (but other, shorter data sets had slightly better correlation in the context of North America). CHIRPS relies on the integration of satellite data and ground observations; this may be problematic in West Africa, where the density of reliable ground-based observations is very sparse; some of these issues are discussed in [22]. Some decision makers in Sierra Leone are already familiar with CHIRPS as its near-real-time estimates are used by VAM [23] for early warning of potential food-security problems due to unusual weather patterns. VAM was used to help mobilize action in Sierra Leone in 2018, when early rains encouraged farmers to plant; the seedlings then died, as the rains became erratic [24]. CHIRPS can be downloaded for free from the internet at a geographic resolution of 0.05 degrees (which equates to just over 5 km in Sierra Leone). CHIRPS data for Africa was downloaded by using an R script called "getCHIRPS" [25]. The coverage for each day was "clipped" to a geographic extent slightly larger than Sierra Leone. Daily data for Sierra Leone and the surrounding area were processed in R [26], to generate different measures of water availability. There are many possible indices that can be used to capture aspects of climate change, [27] discuss 24 (the majority relating to temperature extremes of one sort or another). One index in [27] is the length of the growing season, but this is defined in terms of consecutive days with a temperature above 5 °C, which is of little relevance to Sierra Leone. The length of the rainy season can be estimated directly from the rainfall record; [28] aggregates data into pentads (5-day periods), selects a threshold, and the start is then defined as when the pentad total is above the threshold and the previous pentads are considerably below the threshold (what constitutes "considerably below" is not obvious). A less-arbitrary method is the anomalous accumulation (AA) method [29] illustrated in Equation (1):

$$AA(n) = \sum_{n=1}^{t} \left( R_n - \overline{R} \right) \tag{1}$$

where $R_n$ is the rainfall on day $n$, and $\overline{R}$ is the average daily rainfall. The minimum value of AA(n) corresponds to the start of the rainy season, and the maximum value of AA(n) marks the end of the rainy season.

The measures of water availability below were deemed relevant in the context of Sierra Leone:

- Annual total rainfall (1 January to 31 December).
- Early season rainfall (15 March to 31 May).
- Day number of the start of the main rainy season (the main rainy season being the longest period when the soil moisture content is continuously above 10 mm). In most cases, this is within one week of the start of the rainy season, as estimated by the AA method.
- A measure of how erratic the early rainfall is (number of days between the first day that soil moisture exceeds 10 mm and the start of the main rainy season defined above); this variability is not easily captured with the AA method.

○　Length of the rainy season estimated as the number of "wet" days (more than 10 mm/day).

○　Length of the growing season is estimated as the number of days when the soil moisture content is above 10 mm; this is consistently a month to six weeks after the end of the rainy season, as calculated by the AA method because of the effect of the soil water storage.

○　Extreme events which are characterized as the annual maximum 1-day and annual 7-day maximum rainfall totals. A 5-day max (RX5Day in [27]) is the convention, but farmers in Sierra Leone often talk of 7-day rains and consistently claim they are now much less common.

Early season rainfall during the period 15 March to 31 May is the critical for arable farmers. In 2018, VAM [23] (which uses CHIRPS data) highlighted anomalies in rainfall in March and April. Farmers described the start of the growing season as "erratic" and "deceptive", and a survey by the Food Security Working Group [24] reported 84.9% of households would have to replant some or all of their annual crops.

There is insufficient information to precisely calculate a soil–water balance; in particular, the lack of information on wind speed, incoming solar radiation, and saturation vapor deficits prevents the calculation of evaporation and transpiration on a daily basis and a lack of data on the hydrological properties of soils to calculate water holding capacity and drainage functions. A soil–water balance is approximated by using a simple "leaky bucket" model: Each day, the rainfall is added to the "bucket", and a fixed amount of 2.5 mm "leaks" out (representing a combination of evaporation, transpiration, and groundwater drainage). The bucket has a maximum capacity of 100 mm, which would represent an effective rooting depth of about 600 mm, and an available water holding capacity of about 16%, which is typical of the better agricultural soils in Sierra Leone [30]. Growth of agricultural crops is assumed to occur when there is at least 10 mm of water available. At the start of the rainy season, soil moisture may fluctuate above and below 10 mm until it finally starts to increase to saturation, the difference between the first-time soil moisture exceeds 10 mm and the last time it is below 10 mm before the peak of the rainy season is used as a measure of how erratic the start of the rains has been. The length of the growing season is the longest continuous period with more than 10 mm in the "bucket". This simple model appears slightly more conservative than the method used by the FAO [31] because their map puts most of Sierra Leone in the "Forest Zone", with a growing period greater than 270 days, while our simple model only has three districts (Kenema, Kono, and Kailahun), with a median growing season of 270 days or longer.

Linear regression was used to determine the statistical significance of any trends in the measured rainfall indices. Trends were related to the year and to the maximum monthly temperature anomaly in the Pacific, referred to as El Niño–Southern Oscillation (ENSO). ENSO has a significant effect on precipitation in many places [32,33], but possibly not humid West Africa [4]. The "F-test" was used to try to determine whether the different measures were becoming more or less variable; that is, to test whether the weather is really becoming more erratic. Nonparametric tests such as Kendall's tau were considered, as they make fewer assumptions about statistical distributions and are less sensitive to outliers; but examination of QQ-plots (qqnorm in R) and the Shapiro–Wilks statistic (shapiro.test in R) suggest that there are few if any outliers and the values of the annual total precipitation are normally distributed. The normal distribution provided a better fit to the data than the log-normal or EV (extreme value distribution, such as, Gumbel).

For administrative purposes, Sierra Leone is divided into 14 districts (Figure 1a) (although, in 2016, it was proposed to split two of these districts, and the change is still in progress). The FAO and FEWS-NET (Famine Early Warning Network) have developed a "livelihoods zone" map [34], which summarizes economic activity and resource availability into ten categories. Rice is the critical agricultural product in Sierra Leone, and places in the livelihood zone 10 are termed "rice bowl areas". There are three geographically distinct rice bowl areas, and these correspond to areas where there is the potential for large-scale mechanized rice farming: Coastal sediment (A), Boli swamps (B), and riverine sediment (C) (Figure 1b). For the purpose of this paper, quantities are tabulated by the geographic center of 14 "traditional" (or familiar) districts (pre-2016) and for the three rice bowl areas.

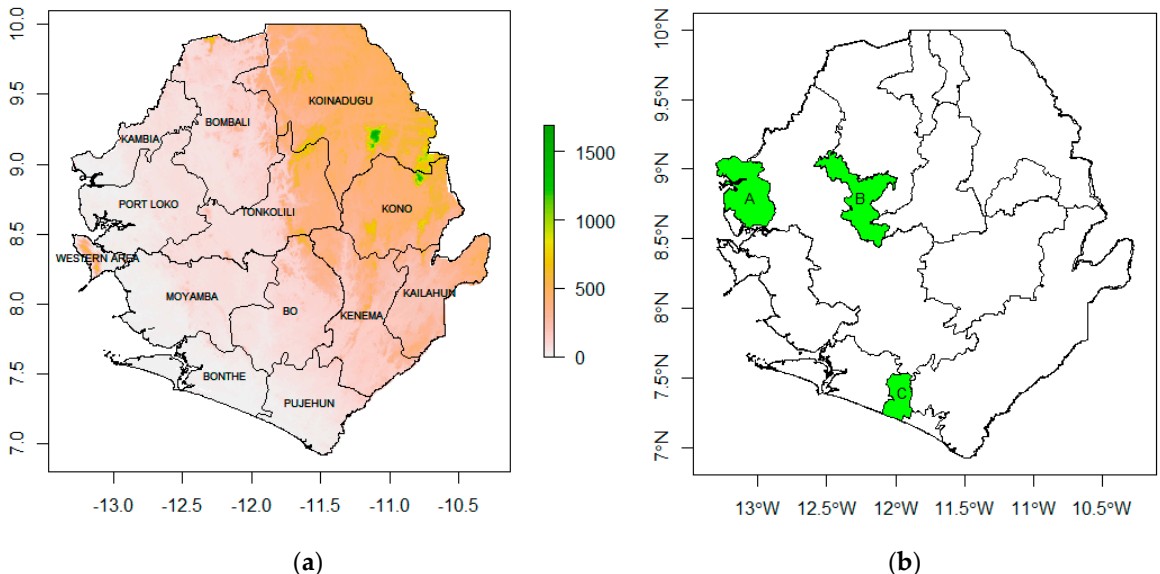

**Figure 1.** Regions of Sierra Leone: (**a**) administrative districts and elevation in meters; (**b**) livelihood zones (the three "rice bowl" areas in green are A—Rokuppr, B—Boli Swamps, and C—Torma Bum).

## 4. Results

### 4.1. Annual Rainfall

Average annual rainfall (1981–2018) is shown in Figure 2. Rainfall is highest along the coastal strip and declines toward the northeast (which is dominated by woody savannah vegetation rather than tropical forest).

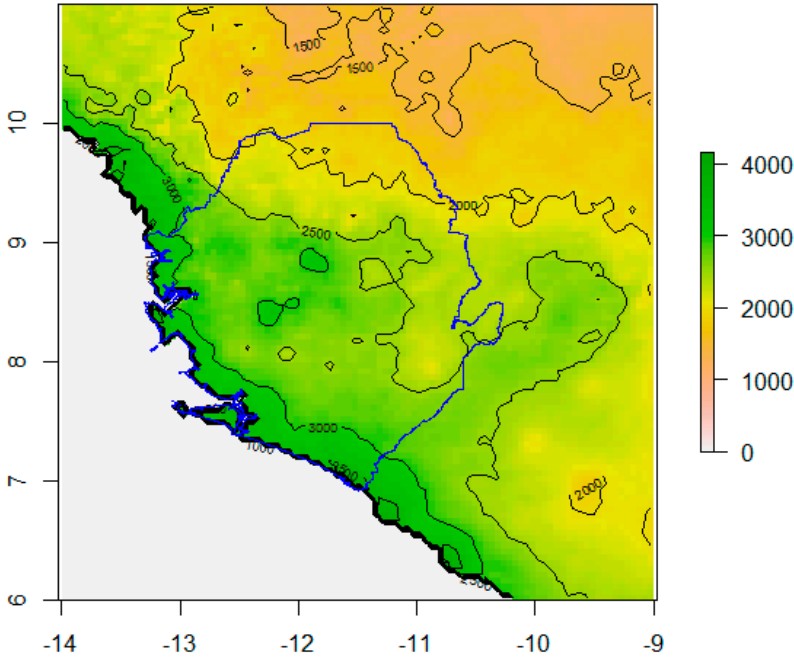

**Figure 2.** Mean average rainfall estimated from Climate Hazards Group InfraRed Precipitation with Station (CHIRPS) daily data. Note the high rainfall area along the coast (>3000 mm/year); only in the far northeast does average annual rainfall fall below 2000 mm/year.

Linear regression of total rainfall against year shows changes of between (a nonsignificant) increase of +0.7 mm/year in Pujehun and (a significant) decline of –15.7 mm/year in the Western

Area (Figure 3a and Table 1). Changes only reach conventional levels of statistical significance in the northwest (Figure 3c and Table 1). The areas where the decline is statistically significant include one of the rice bowl areas (A on Figure 1b) and the capital (Freetown), which is by far the largest metropolitan area of the country and is home to nearly one-third of the population.

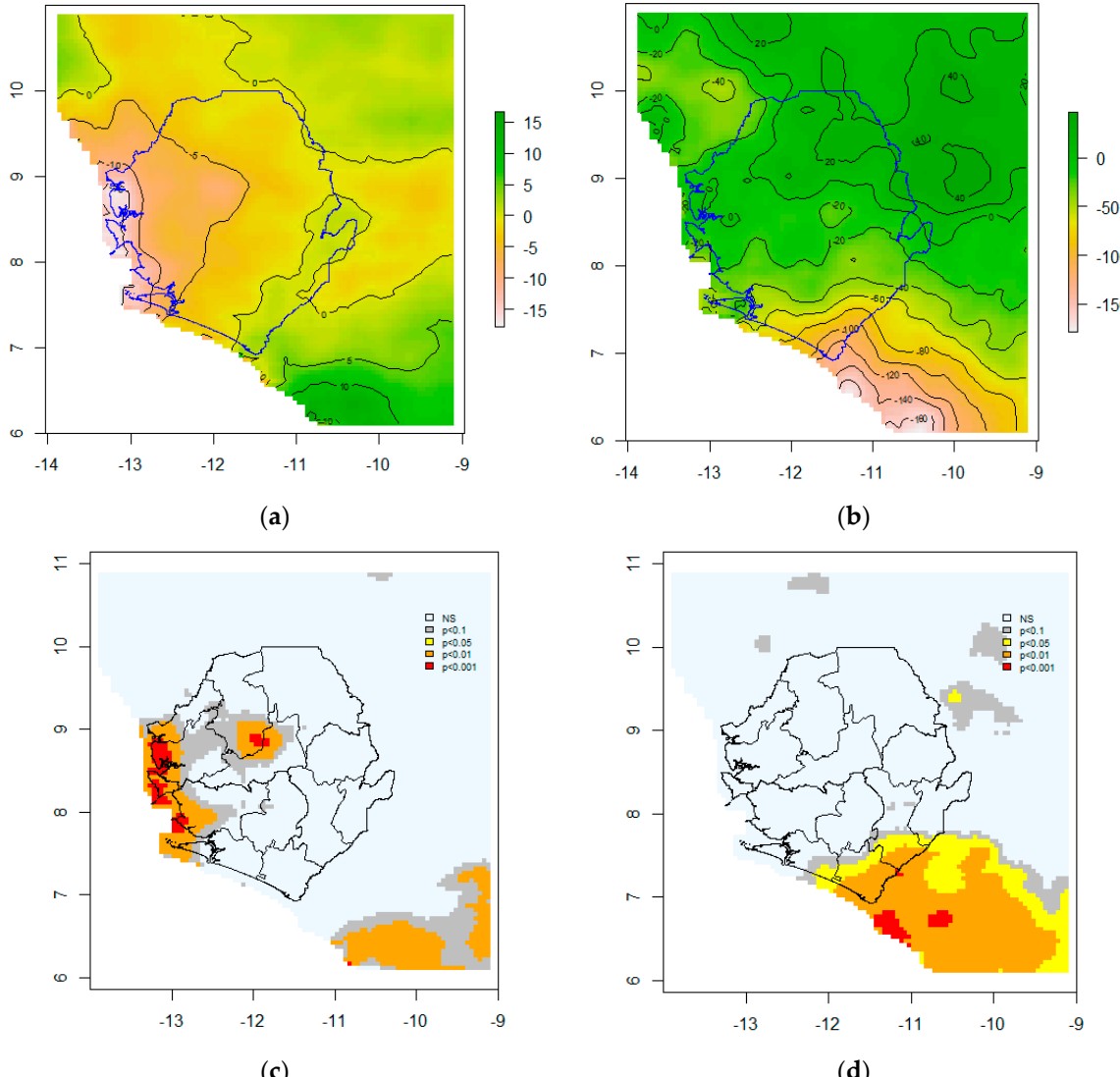

**Figure 3.** Trends in total annual rainfall: (**a**) trend in total rainfall with time, (**b**) impact of a 1 °C temperature anomaly in the Pacific (El Niño) on annual rainfall, (**c**) significance of the annual trend with time, and (**d**) significance of the effect of El Niño.

There is a statistically significant relationship between the median annual rainfall and the predicted rate of decline (adjusted $r^2 = 0.287$, df = 13, $p < 0.05$) (see Figure 4), but note that the variance increases with the rainfall, pointing to less-dependable regression results.

**Table 1.** Total annual rainfall 1981–2018.

| Region | Median Rainfall mm | Trend with Time | | | Trend with El Niño | |
|---|---|---|---|---|---|---|
| | | mm/1981–2018 | Prob' | Variability | mm/+1 °C | Prob' |
| Kambia | 2755 | −259 | 0.129 | Inc | −5.2 | 0.897 |
| Bombali | 2511 | −195 | 0.094 * | Inc | 13.8 | 0.617 |
| Koinadugu | 1987 | −81 | 0.358 | Dec | 25.3 | 0.215 |
| PortLoko | 2788 | −305 | 0.090 + | Inc | −2.9 | 0.946 |
| Tonkolili | 2896 | −278 | 0.034 * | Dec | −9.1 | 0.773 |
| Kono | 2288 | −67 | 0.452 | Dec * | −0.8 | 0.968 |
| WA | 3471 | −598 | 0.010 ** | Dec | −17.9 | 0.750 |
| Moyamba | 2766 | −260 | 0.066 + | Inc | −26.0 | 0.438 |
| Bo | 2574 | −162 | 0.113 | Dec | −19.7 | 0.413 |
| Kenema | 2540 | −25 | 0.826 | Dec | −38.6 | 0.130 |
| Kailahun | 2338 | 28 | 0.748 | Dec | −26.9 | 0.187 |
| Bonthe | 3232 | −155 | 0.353 | Dec | −56.5 | 0.140 |
| Pujehun | 3129 | 8 | 0.960 | Dec | −84.8 | 0.011 + |
| RB_A | 3126 | −605 | 0.009 ** | Dec | −0.6 | 0.991 |
| RB_B | 2745 | −233 | 0.093 + | Inc | −12.1 | 0.712 |
| RB_C | 3111 | −73 | 0.620 | Dec | −61.5 | 0.068 + |

Notes: Values are for the geographical midpoint of each region. Trend with time and El Niño from linear regression, change in variability from the F-test. Variability Inc = increasing, Dec = decreasing. Significance levels; + $p < 0.1$, * $p < 0.05$, ** $p < 0.01$. RB_A Rokuppr (coastal sediment), RB_B Boli swamps, and RB_C Torma Bum (riverine grassland).

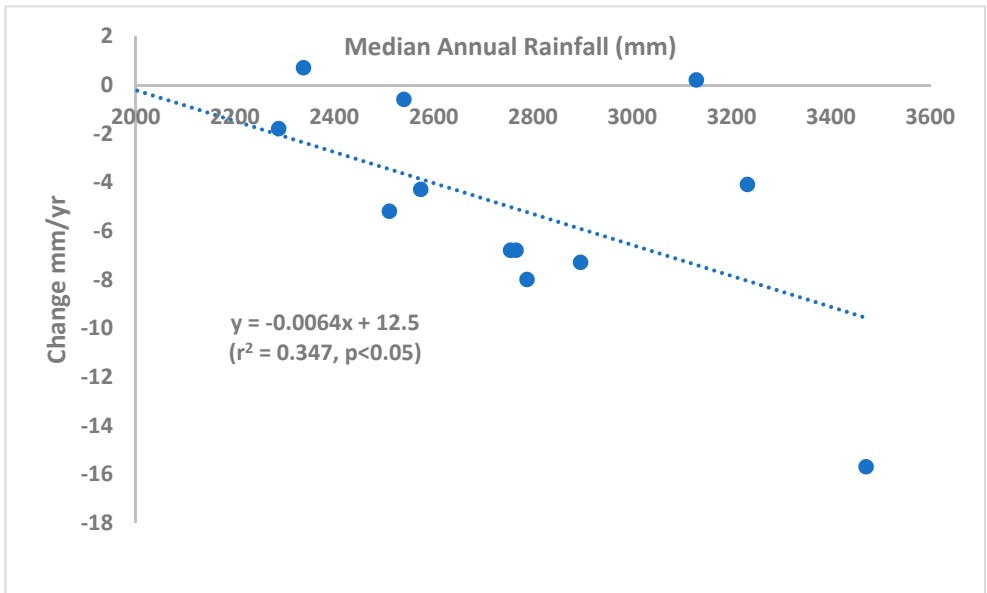

**Figure 4.** Relation between median rainfall (mm) in each district and the rate of change in annual rainfall (mm/yr).

Annual rainfall in some parts of the world is affected by El Niño events [35]. In Sierra Leone, there is an effect which becomes statistically significant in Pujehun District in the southeast of the country (Figure 3b,d and Table 1). Rainfall totals were correlated with the maximum temperature anomaly of the preceding year (shown for Pujehun in Figure 5). Annual rainfall declines with positive temperature anomalies in the Pacific ("El Niño") and increases with negative anomalies ("La Nina"). There is some indication that El Niño events may become stronger and more common [36]; whether this occurs, if they can be predicted at least a few months in advance, then there is some possibility of warning of unusual weather to come in that season.

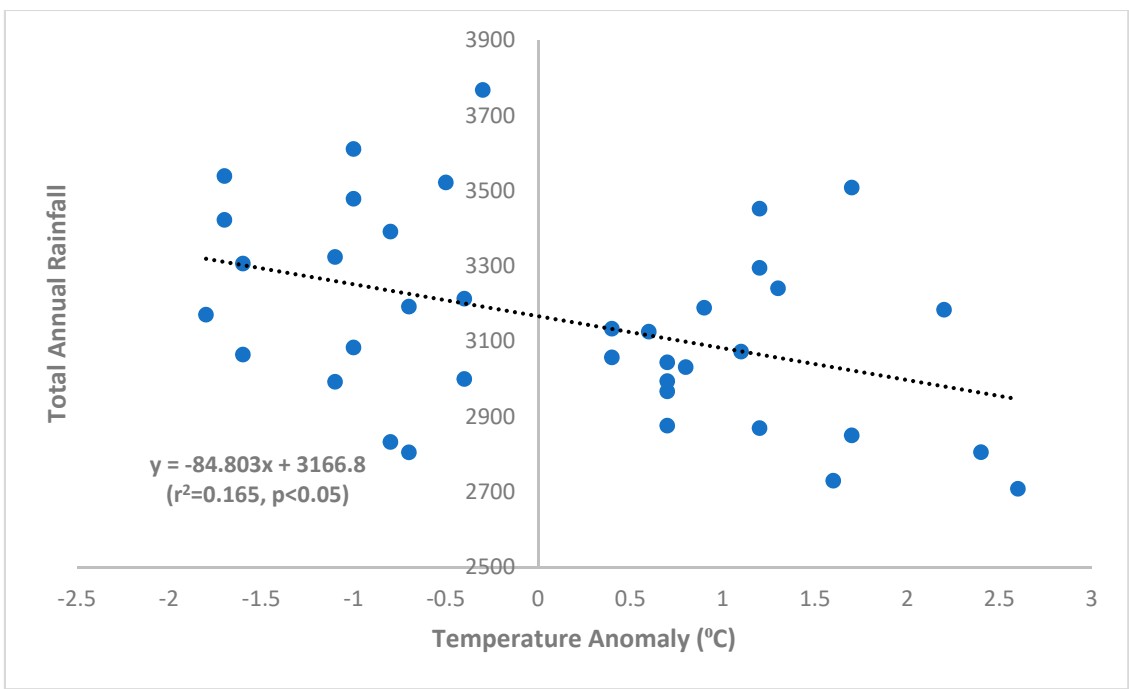

**Figure 5.** The effect of the El Niño southern oscillation temperature anomaly on annual rainfall in Pujehun.

The variance in rainfall was calculated for the period before and after the year 2000 (which bisects the time period), and an "F-test" was performed to see whether the variability of the total rainfall was different in the two time periods (Table 1). Results were mixed, with no clear pattern of increasing or decreasing variability; the majority showed a decrease in the CV (coefficient of variation), but only in Kono were the results statistically significant.

*4.2. Early Season Rainfall*

The period from the middle of March to the end of May is critical for farmers. If they plant too soon, there is the risk of a dry period and seedlings dying; and if they wait too long, weeds can become established, and newly emerged seedlings can be damaged by heavy rain. The early season rainfall shows consistent declines across the whole country (Figure 6a,b and Table 2). The same general pattern that was seen in the total annual rainfall is observed with a decline everywhere, but it is statistically significant only in a small coastal band. Early season rainfall declines everywhere during El Niño events but is nowhere statistically significant.

The variance in rainfall was calculated for each decade, and an "F-test" was performed to test whether the variability of the early season rainfall was increasing over time. Results show that the variability in the early rainy season total is decreasing in most regions (as the total amount also decreases). Variability increased in only two districts (Bonthe and Moyamba) but significantly declined in Kono and Kailahun (Table 2).

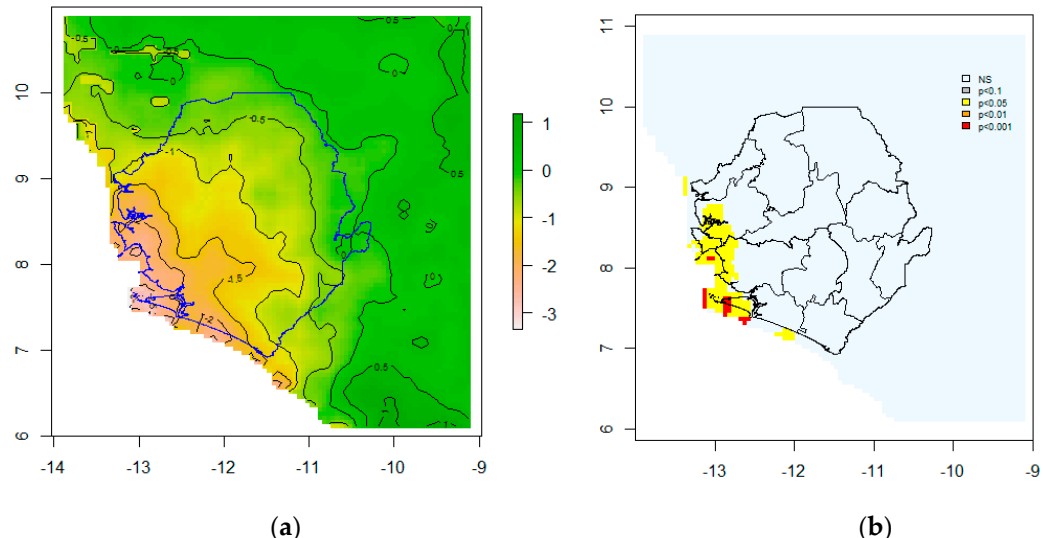

(**a**)　　　　　　　　　　　　　　　　　　　　　　　(**b**)

**Figure 6.** Early Season rainfall (15 March to 31 May): (**a**) changes in rainfall in mm/year; (**b**) statistical significance of the change in early season rainfall.

**Table 2.** Early season rainfall (15 March to 30 May).

| Region | Median Value mm | Trend over Time | | | Trend with El Niño | |
|---|---|---|---|---|---|---|
| | | mm/yr | Prob′ | Variability | mm/°C | Prob′ |
| Kambia | 184 | −1.17 | 0.257 | Dec | −1.65 | 0.858 |
| Bombali | 232 | −0.89 | 0.374 | Dec | −2.08 | 0.815 |
| Koinadugu | 247 | −0.58 | 0.557 | Dec | −2.58 | 0.770 |
| PortLoko | 216 | −1.45 | 0.169 | Dec | −1.85 | 0.844 |
| Tonkolili | 298 | −1.12 | 0.345 | Dec | −7.33 | 0.485 |
| Kono | 348 | −0.47 | 0.676 | Dec * | −5.06 | 0.608 |
| WA | 226 | −2.03 | 0.103 | Dec | −5.17 | 0.644 |
| Moyamba | 279 | −1.63 | 0.196 | Inc | −9.57 | 0.395 |
| Bo | 311 | −1.49 | 0.214 | Dec + | −5.19 | 0.628 |
| Kenema | 345 | −1.18 | 0.414 | Dec | −8.09 | 0.528 |
| Kailahun | 376 | −0.10 | 0.936 | Dec * | −4.84 | 0.661 |
| Bonthe | 295 | −1.87 | 0.171 | Inc | −14.02 | 0.246 |
| Pujehun | 362 | −1.02 | 0.457 | Dec + | −12.40 | 0.305 |
| RB_A | 186 | −1.71 | 0.114 | Dec | −2.64 | 0.786 |
| RB_B | 237 | −1.11 | 0.287 | Dec | −3.58 | 0.699 |
| RB_C | 301 | −1.60 | 0.210 | Dec | −10.19 | 0.368 |

Notes: Values are for the geographical midpoint of each region. Trend with time and El Niño from linear regression, change in variability from the F-test. Variability Inc = increasing, Dec = decreasing. Significance levels; + $p < 0.1$, * $p < 0.05$. RB_A Rokuppr (coastal sediment), RB_B Boli swamps, and RB_C Torma Bum (riverine grassland).

*4.3. Start of the Rainy Season*

Soil moisture is estimated by using a simple leaky bucket model. Agricultural crop growth is assumed to start once soil moisture is equivalent to 10 mm of water. In some years, soil moisture fluctuates around the 10 mm threshold before it finally starts a substantial increase of the main rainy season. The number of days between the first-time soil moisture exceeds 10 mm and the last day it is less than 10 mm (before the peak of the rains) is an indication of how erratic the early rainy season is. Table 3 shows the median value for the day number when the main rainy season starts; in all but two districts (Kono and Kailahun), the rainy season has been starting later (median value of 11.6 days for all the districts over the entire period of record). An F-test suggests that the variability has been increasing in a few districts, mostly in the north and west.

**Table 3.** The rainy season (day number).

| Region | Median Day | Change over Time | | |
| --- | --- | --- | --- | --- |
| | | **Change 1981–2018** | **Prob'** | **Variability** |
| Kambia | 124 | 11.0 | 0.240 | Inc |
| Bombali | 117 | 8.1 | 0.177 | Dec + |
| Koinadugu | 118 | 11.6 | 0.221 | Dec + |
| PortLoko | 119 | 9.3 | 0.201 | Dec * |
| Tonkolili | 109 | 14.7 | 0.294 | Inc * |
| Kono | 89 | −2.6 | −0.040 * | Inc |
| WA | 118 | 5.0 | 0.109 | Dec * |
| Moyamba | 117 | 3.9 | 0.079 + | Dec + |
| Bo | 111 | 15.1 | 0.335 | Dec |
| Kenema | 97 | 21.2 | 0.306 | Inc |
| Kailahun | 86 | −1.2 | −0.022 * | Inc |
| Bonthe | 116 | 14.0 | 0.239 | Inc |
| Pujehun | 106 | 15.9 | 0.352 | Inc |
| RB_A | 124 | 12.2 | 0.260 | Inc |
| RB_B | 119 | 10.4 | 0.220 | Dec |
| RB_C | 113 | 12.8 | 0.238 | Dec |

Notes: Values are for the geographical midpoint of each region. Trend with time from linear regression, change in variability from the F-test. Variability Inc = increasing, Dec = decreasing. Significance levels; + $p < 0.1$, * $p < 0.05$. RB_A Rokuppr, RB_B Boli swamps, and RB_C Torma Bum.

Table 3 shows the variability in the start of the rainy season has been assessed by estimating the number of days between the first day in the year when soil moisture exceeds 10 mm and the last time day in which it is below 10 mm before mid-August (the peak of the rains). There is no clear pattern that shows whether the period between the first- and last-time soil moisture exceeds 10 mm at the start of the rainy season is changing.

*4.4. Number of Days with Rain Greater than 10 mm and Length of the Growing Season*

The length of the rainy season is approximated as a simple count of the number of days with rainfall greater than 10 mm. Figure 7a,d and Table 4 indicate that the "length" of the wet season is in decline in every district, with the largest declines being in the north and west, and a statistically significant decline being in Bombali District.

Number of days with rainfall greater than 10 mm declines during El Niño events, with the largest effect in Pujehun, Bo, Kenema, and Bonthe Districts (south). Using a threshold of 5 mm gives a similar pattern, (not shown) the differences between the number of days with 5 and 10 mm of rain are small. The variability of the length of the wet season seems to be increasing slightly in the north and declining slightly in the south (Table 4).

Length of the growing season is estimated as the longest continuous period when the soil moisture content (modelled by a "leaky bucket") is above 10 mm. Table 4 shows the median length of the growing season and the trend over time. There are very minor changes over the period of study. There is, however, a very strong correlation in all districts ($p < 0.001$) between when the rainy season starts and how long it lasts; when the rainy season starts late, it will be short.

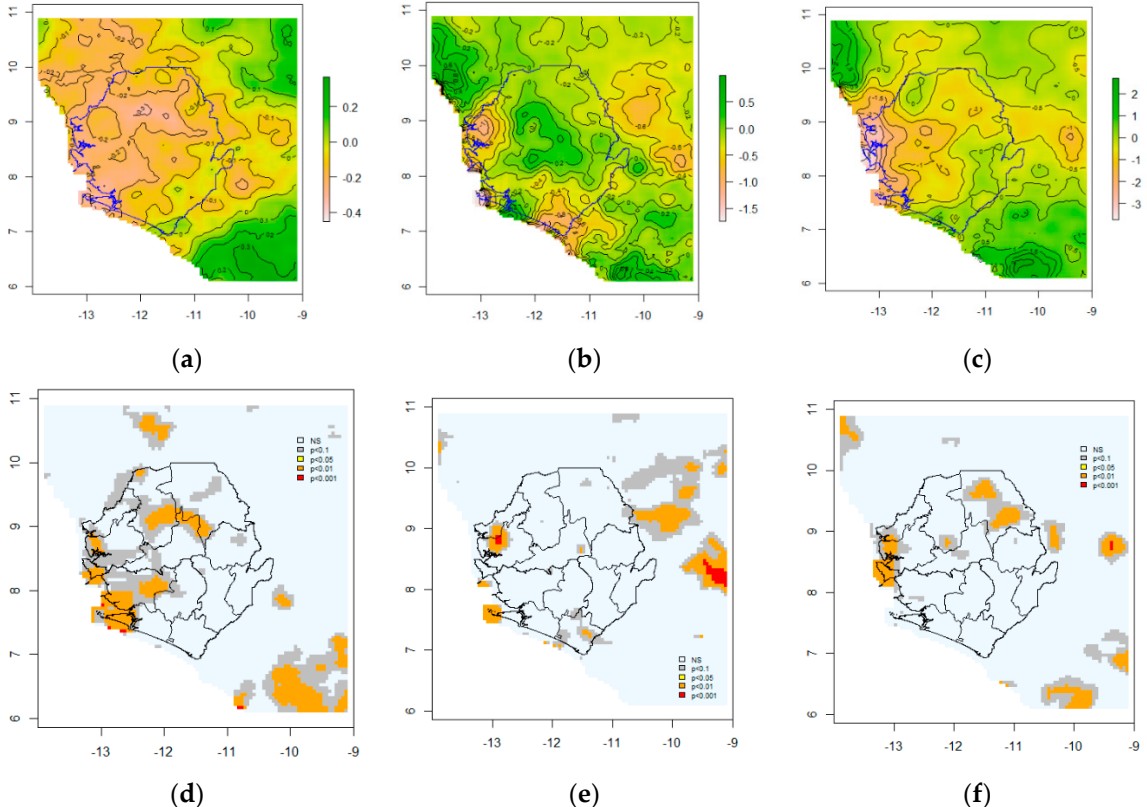

**Figure 7.** Wet days and extreme events: (**a**) trend in the number of days with rain greater than 10 mm (days/years); (**b**) trend in the annual maximum one-day rainfall event (mm/year); (**c**) trend in annual maximum seven-day rainfall event (mm/year). (**d**) Significance of the trend in the number of days with rainfall greater than 10 mm. (**e**) Significance of the trend in the annual maximum one-day rainfall event. (**f**) Significance of the trend in the annual maximum seven-day rainfall event.

**Table 4.** The length of the wet season with time and El Niño (days with >10 mm of rain and length of the growing season).

| Region | Number of Days with Rain >10 mm | | | | Growing Season (Days) | | | |
|---|---|---|---|---|---|---|---|---|
| | Change | | | | Change | | | |
| | Median | Days/Year | Days/°C | Variability | Median | Days/Year | Days/°C | Variability |
| Kambia | 90 | −0.17 | −1.40 | Inc * | 240 | −0.20 | −0.13 | Dec |
| Bombali | 88 | −0.29 * | −0.75 | Inc | 240 | −0.12 | 2.34 | Dec |
| Koinadugu | 78 | −0.21 + | −0.18 | Dec | 243 | −0.07 | −1.76 | Inc |
| PortLoko | 92 | −0.20 | −1.47 | Inc * | 245 | −0.13 | −0.12 | Dec |
| Tonkolili | 97 | −0.19 | −1.62 | Dec | 244 | −0.40 + | 1.41 | Dec + |
| Kono | 94 | −0.17 | −0.42 | Dec * | 260 | 0.18 | −2.37 | Inc |
| WA | 94 | −0.25 + | −1.92 | Dec | 274 | −0.13 | 1.73 | Dec |
| Moyamba | 91 | −0.25 + | −2.06 + | Dec + | 247 | −0.03 | 0.78 | Dec |
| Bo | 89 | −0.20 + | −1.11 | Dec + | 251 | −0.27 + | 0.91 | Dec + |
| Kenema | 90 | −0.06 | −2.18 * | Dec | 258 | −0.43 + | −1.76 | Dec + |
| Kailahun | 91 | 0.00 | −0.82 | Dec * | 270 | 0.12 | −1.37 | Inc |
| Bonthe | 90 | −0.14 | −2.71 * | Dec ** | 282 | −0.37 + | 2.84 | Dec + |
| Pujehun | 94 | −0.10 | −3.78 ** | Dec * | 252 | −0.41 * | −0.86 | Dec * |
| RB_A | 94 | −0.25 * | −1.55 | Inc + | 265 | −0.15 | −1.48 | Dec |
| RB_B | 93 | −0.21 | −1.24 | Inc + | 239 | −0.19 | −1.15 | Dec |
| RB_C | 91 | −0.13 | −2.45 * | Dec ** | 245 | −0.25 | 1.55 | Dec |

Notes: Values are for the geographical midpoint of each region. Trend with time and El Niño from linear regression, change in variability from the F-test. Variability Inc = increasing, Dec = decreasing. Significance levels; + $p < 0.1$, * $p < 0.05$, ** $p < 0.01$. RB_A Rokuprr, RB_B Boli swamps, and RB_C Torma Bum.

*4.5. Peak One-Day Rainfall and Seven-Day Rainfall*

The estimated change in the annual maximum one-day rainfall is shown in Figure 7b,e. The largest values are observed in the Western Area (median value 154 mm) and lowest in Koinadugu (median value 59.3 mm). Within all districts, the largest observed annual one-day event is always at least twice the size of the smallest annual one-day event (for example, in the Western Area in 1995, the largest one-day event was 85.1 mm, but in 1982, the largest one-day event was 248 mm; similarly, in Koinadugu in 1994, the largest one-day event was 93 mm, but it was only 40 mm in 1984). There are no significant trends with time or with El Niño events (Table 5). Erosion and flooding are both strongly influenced by topography as well as ground cover, and reports of an increase in flooding and landslides, particularly in Freetown, seem more likely a result of the clearing of forested slopes for housing rather than an increase in the intensity of one-day events.

The variability in the maximum one-day rainfall event is decreasing in most districts and is statistically significant in half of the districts (Table 5).

The pattern of maximum seven-day rainfall is close to that of the one-day maximum (Figure 7c,f and Table 5). Changes in variability of the annual maximum seven-day event are mixed (Table 5), with an increase in some districts and a decline in others.

**Table 5.** Extreme events; one-day and seven-day rainfall totals.

| Region | One-Day Totals | | | | Seven-Day Totals | | | |
|---|---|---|---|---|---|---|---|---|
| | **Median mm** | **Change** | | | **Median mm** | **Change** | | |
| | | **mm/yr** | **mm/°C** | **Variability** | | **mm/yr** | **mm/°C** | **Variability** |
| Kambia | 188.3 | −0.51 | 2.21 | Dec * | 311 | −1.1 | 10.4 | Inc |
| Bombali | 134.3 | 0.31 | −0.20 | Inc * | 261 | −0.6 | 3.8 | Inc +. |
| Koinadugu | 93.1 | −0.14 | 0.76 | Dec + | 183 | −0.7 + | 1.2 | Dec + |
| PortLoko | 183.8 | −0.57 | 2.84 | Dec * | 306 | −1.9 | 11.1 | Dec + |
| Tonkolili | 152.0 | 0.14 | −0.56 | Dec | 305 | −1.4 + | 8.3 | Dec + |
| Kono | 109.5 | 0.05 | −0.04 | Dec | 195 | −0.4 | −0.8 | Inc + |
| WA | 248.3 | −0.56 | 6.14 | Dec | 399 | −3.5 * | 9.2 | Dec ** |
| Moyamba | 162.6 | −0.24 | 2.12 | Inc | 287 | −1.3 | 6.9 | Dec |
| Bo | 143.5 | −0.26 | 1.72 | Dec * | 244 | −1.0 | 2.9 | Inc |
| Kenema | 176.2 | −0.21 | −1.43 | Dec *** | 239 | 0.0 | −0.2 | Inc |
| Kailahun | 130.3 | −0.17 | −0.77 | Dec * | 198 | −0.3 | 0.5 | Inc |
| Bonthe | 217.4 | 0.02 | 0.56 | Dec | 358 | −0.6 | 4.3 | Inc |
| Pujehun | 245.2 | −0.86 + | 2.73 | Dec *** | 325 | 0.0 | −2.5 | Inc |
| RB_A | 239.2 | −0.95 | 2.12 | Dec* | 407 | −3.0 * | 9.1 | Dec + |
| RB_B | 160.3 | 0.22 | 0.59 | Dec | 284 | −1.3 | 7.3 | Dec |
| RB_C | 233.3 | −0.29 | 4.48 | Dec ** | 327 | −0.1 | 2.2 | Inc * |

Notes: Values are for the geographical midpoint of each region. Trend with time and El Niño from linear regression, change in variability from the F-test. Variability Inc = increasing, Dec = decreasing. Significance levels; + $p < 0.1$, * $p < 0.05$, ** $p < 0.01$, *** $p < 0.001$. RB_A Rokuppr (coastal sediment), RB_B Boli swamps, and RB_C Torma Bum (riverine grassland).

## 5. Discussion

There is considerable concern that Sierra Leone is very vulnerable to the economic impacts of climate change (one company ranks it as the third most vulnerable in the world [9]). Unfortunately, the plethora of climate models provides an extremely wide range of predictions, so it is difficult to justify any statement other than "rainfall is probably going to change". There are very few reliable long-term meteorological records available, and instead the CHIRPS data set is used because it has a long record, a high spatial resolution, is already familiar to some decision makers in Sierra Leone, and is in good agreement qualitatively with observed changes in phenology, such as the erratic start to the 2018 planting season. The World Bank Climate Change Data portal [14] provides monthly data from 1901 to 2015 (provided by the CRU, University of East Anglia). Linear regression shows a statistically significant decline in annual rainfall of 2.8 mm/year (adjusted $r^2 = 0.171$, $p < 0.001$, df = 114); this is about 60% of the rate of decline estimated by CHIRPS over the last four decades. The strongest

correlation between the World Bank annual totals is with the CHIRPS data for Kenema; unfortunately, this is not quite statistically significant ($p = 0.091$, df = 34, adjusted $r^2 = 0.054$), nor is there a statistically significant correlation with any of the four stations reported by Statistics Sierra Leone [16]. The stations in the Statistical Digest are not significantly correlated with each other, even for Freetown and Lungi, where the sites are less than 20 km apart. A fourth station in the Statistical Digest records an annual rainfall about one meter less than might be expected from other data.

Using the CHIRPS data, there are significant declines in the north and west in total annual rainfall and, to a lesser extent, to early rainy season rainfall and length of the growing season. In the south and east, there is less of a change with time; instead, annual rainfall responds to El Niño and La Nina events in the Pacific.

There is no indication in the data that extreme events in the form of annual maximum one-day and seven-day totals are becoming more common, nor do extreme events appear to be related to El Niño events. In Brazil, flooding and landslides were observed to be increasing [37], even though extreme events were not statistically significant, so a link seems plausible. The increased reports of flooding, landslides, and erosion seem more likely to be attributed to anthropogenic factors, in the form of urban expansion onto previously forested hillsides in Freetown [38], rather than an increase in extreme events. In Brazil, an increase in dry spells and a later start to the rainy season was associated with increased levels of deforestation [39]. The rate and timing of deforestation is a contested topic in Sierra Leone [40], although it is clearly visible in and around Freetown.

Analysis of variability in all measures of water availability (using the F-test) to see whether variance is significantly higher or lower in the second half of the record (pre and post the year 2000), show a reduction in variability in many cases.

## 6. Conclusions

Sierra Leone has a history of conflict that may, at least in part, be attributed to conflict over natural resources [1]; it is also considered to be exceptionally sensitive to the economic effects of climate change [9]. Changes in rainfall, extreme events, or the length of the growing season could adversely affect the economy and reignite some of these tensions; it is critical that an objective and realistic assessment of the likelihood of change is available to decision makers and policy makers. There is a statistically significant reduction in annual rainfall is in the northwest; this coincides with one of the "rice bowl" areas and with the largest metropolitan area. If these trends in rainfall continue, they will have a detrimental impact and a synergistic effect with increased temperatures and deforestation of the steep hillsides characteristic of the area, leading to erosion, the silting up of reservoirs, landslides, the drying up of steams, and other environmental problems. Despite high rainfall, potable water is already a major problem in the capital Freetown, as Guma reservoir was designed to supply 600,000, but the population in the greater metropolitan area is now approaching two million and is rapidly growing with rural–urban migration. The most affected "rice bowl" area (A) is situated on what was originally mangrove swamps. The soils are dominated by marine alluvium, and successful cultivation of rice depends on high rainfall to reduce salinity, so a significant reduction in rainfall will adversely affect this area. Many other crops may be affected, directly and indirectly, through reduced pollination or acute water stress at fruit set or seed maturity. The future role of research institutions and universities ought to be to test possible adaptive strategies now, before the effects become significant everywhere.

**Author Contributions:** Conceptualization, R.W. and A.L.; methodology, R.W.; software, R.W.; validation, R.W., A.J., formal analysis, R.W.; writing—Original draft preparation, R.W.; writing—Review and editing, A.J. and A.L..; visualization, R.W.; supervision, A.L.

**Funding:** This research and the APC was funded by UPGRADE Plus, grant number BLE—2816PROC01. UPGRADE Plus project is supported by funds of the Federal Ministry of Food and Agriculture (BMEL) via the Federal Office for Agriculture and Food (BLE).

**Acknowledgments:** We wish to thank the creators of CHIRPS (Climate Hazards Group InfraRed Precipitation with Station data) for making their dataset available. We wish to thank Amber Martin (University of Illinois) for checking the English and proofreading the manuscript.

**Conflicts of Interest:** The authors declare no conflicts of interest.

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
