# Peer review of "Changes in Rainfall in Sierra Leone: 1981–2018"

_climate, doi:10.3390/cli7120144_

Round 1

Reviewer 1 Report

Review of “Changes in Rainfall in Sierra Leone: 1981-2018” by Richard Wadsworth, Amiee Jalloh, and Aiah Lebbie.

The authors present a detailed look at changes in rainfall characteristics for Sierra Leone over the past 4 decades. Given limitations in observational data, the authors utilize the CHIRPS precipitation dataset, which is archived at a high resolution. The authors examine changes in annual rainfall amount, daily rainfall intensity, daily rainfall variability, rainy season onset and length, and relationships between rainfall and ENSO.  They find evidence for overall drying, with statistically significant changes found in the north and west of the country. Extreme precipitation in the form of heavy rainfall does not exhibit statistically significant changes during the study period.

Overall, I think this is a welcome contribution to the collection of scientists and stakeholders interested in Sierra Leone precipitation.   However, I have several suggestions I think would improve the paper.

The paper is lacking a description of the ENSO index used. Please include it for clarity. The paper is lacking a description of the limitations of the CHIRPS dataset. In your abstract, you should remove the line “such models are complex”. The fact that they are complex isn’t “unfortunate”. Instead, you could say that cause and effect in the models can be difficult to interpret. Line 98 “How erratic the start of the rainy season is?”. You should remove the “?”. In Table 1, indicate what the “°”, “*” and “**” represent in the Table caption. In addition to Figure 1, it would be interesting to plot the percent change in precipitation (y-axis) vs the median annual rainfall (x-axis). How might increasing temperatures influence your simple soil moisture bucket model calculation? Please include that information.

Author Response

Dear Reviewer 1,

Thank you for you many thoughtful suggestions. We hope that the revised manuscript meets your concerns.

In particular we have;

revised the section where we talk about ENSO.

revised the language

inserted explicitly the meaning of *, ** and *** in the column of statisticall significance.

the rise in temperature will increase how "leaky" our "leaky bucket" is, but without more detailed information than we currently have, it is not possible to quantify the magnitude of the change.

We thank you for your time and comments.

Regards

Richard Wadsworth

Reviewer 2 Report

In my opinion, the idea of the paper is worth of presentation. However, there are many things needed to improve the paper.

I would suggest to reconsider English language. Especially at the Introduction I would suggest not to use " previous studies paragraphs". In my opinion all the "Introduction" section should be re-written. Additionally,  there are many syntax errors.

I would also suggest to avoid "we" and use the third person instead.

Some other corrections:

Lines 109-112: Please Correct syntax.

Line 112:getCHIRPS?

Lines 134-138: Please Correct syntax.

Lines 155-170. I think that this section should not belong to the Results.

Lines 208-210: Please Correct syntax.

Line 229: Please correct "were the result".

Lines 258-260.Please Correct syntax.

Lines 266-267: Please Correct syntax.

Lines 277-278: Please Correct syntax.

Lines 294-296: Please Correct syntax.

Lines 370-372: Please Correct syntax.

Lines 374-378: Please Correct syntax.

Finally I would suggest to split "Discussions/Conclusions".

Author Response

Dear Referee 2

Thank you for your comments, which were useful in helping us improve the manuscript.

Please see revised document with "track changes" switched on.

The introduction has been re-written.

All uses of the word "we" have been removed, except for those in direct quotes from other authors and in the acknowledgements.

The syntax has been changed in all the sections identified from 109-112 to 374-378; and we have asked Amber Martin (of the University of Illinois) to check the language.

The discusion/conclusion has been split into two as you suggested.

Thank you again for your comments.

Regards

Richard Wadsworth

Reviewer 3 Report

Dear Editors and Ms.Datura Yang Assistant Editor of MDPI journals,

My revised version of the paper “Changes in Rainfall in Sierra Leone: 1981-2018” has been revised according to manuscript number Climate-637508.

 SOME DETAILED COMMENTS (MAJOR REVISIONS):

 In this research, the authors studied the variability and changes of rainfall over Sierra Leone during 1981-2018. The paper fits within the stated scope of the journal for Climate; however, the manuscript should be improved mainly before considering for publication, specifically in the introduction, methodology, results, and conclusions sections. For instance, the abstract and conclusions parts are a bit confusing because it does not report direct and clear results.  

The second important point is the authors not provide logic explanations of methods and data used. The authors do not offer an appropriate literature review.

The third point is about the presentation of Maps, Figures, and Tables. All figures and tables are not appropriate and legible, and the legend of some figures have annotations that are part of the results (e.g., Map 6). The authors need panels to show a large number of similar figures, and in my opinion, all figures should call “figures”, not maps. My concern here is the figures not offers to the lector clear information to understand the main results.

Finally, the authors need to show what is the novelty and the contributions of the study this study in Sierra Leone concerning the other studies in the world. I consider that this paper is not very clear or compelling enough for publishing.

Thanks,

Reviewer.

Author Response

Dear Referee 3,

Thank you for your comments, which, we hope have greatly improved the manuscript. In particular we would like to thank you for pointing out several references that had escaped our attention.

The historical quotes have been reduced and the reason for including them, has, we hope, been made more explicit. The issue is that in the past influential decision makers have become convinced that something is happening baased on limited and perhaps ultimately misleading data. The same thing seems to be happening now and this paper is in part to try and provide some quantitative evidence for what has happened. There does appear to be a statistically significant reduction in rainfall in the north-west of the country and a non-statistically significant trend in the same direction in the rest of the country; while this is worrying, it needs to be put in the context of other issues such as rural-urban migration and landscape changes.

For the particular suggestions we:

have changed maps to figures

revised the abstract

included many of the references you suggested that had escaped our attention, particularly Beck et al (a very interesting paper)

rewritten the introduction removing many of the historical quotes

revised the methodology

expanded the section detailing why CHIRPS was used

rewritten all sections

asked Amber Martin (University of Illinois) to check the English used in the manuscript.

Thank you again for your insightful comments.

Regards

Richard

Round 2

Reviewer 2 Report

I think that the manuscript has to be re-submitted because has many typographic errors and it is not acceptable.

Thank you.

Author Response

Dear Reviewer 2,

We have revised and simplified the language in the introduction. In case one of the issues was a difference between English-English and American-English I have asked a collegue who was copy editor for a Legal Journal in Illinois (USA) to check for gramatical and other issues.

Regards

Richard

Reviewer 3 Report

Dear Editors and Ms.Datura Yang Assistant Editor of MDPI journals,

I attached the second revised version of the paper “Changes in Rainfall in Sierra Leone: 1981-2018” has been revised according to manuscript number Climate-637508. After the following change that I asked at the first revision, the paper will be ready for publishing.

TABLES AND FIGURES (Important Changes)

In the first revised version, one of the critical “major revision” was “The tables and figures can be joined to summarize the information as panels and provide more details about statistical features”. According to the new version of the MS no changes were done concerning this. Please show the following example and try to reduce the number of FIGURES and TABLES. For example, the authors need panels to show a large number of similar figures

See the example (pdf)

Cheers,

Reviewer. 

Author Response

Dear Reviewer 3,

Thank you for your efforts in getting this paper into a publishable form. In light of your suggestions we have merged the 13 tables into 5 and reduced the number of figures from 17 to 7.

Yours sincerely

Richard Wadsworth